# Angiographic Characteristics and Outcomes of Percutaneous Coronary Intervention of Reattempted Chronic Total Occlusion: Potential Contributing Factors to Procedural Success

**DOI:** 10.3390/jcm10235661

**Published:** 2021-11-30

**Authors:** Mohsen Mohandes, Cristina Moreno, Mónica Fuertes, Sergio Rojas, Alberto Pernigotti, Diego Zambrano, Marta Guillén, Jordi Guarinos, Alfredo Bardají

**Affiliations:** Interventional Cardiology Unit, Cardiology Division, Joan XXIII University Hospital, Pere Virgili Health Research Institute (IISPV), 43005 Tarragona, Spain; crystynama@hotmail.com (C.M.); mon_fuertes@yahoo.com (M.F.); serchmed7@hotmail.com (S.R.); a.pernigotti@gmail.com (A.P.); DIEGOANDRESZ@hotmail.com (D.Z.); sibawayh_82@hotmail.com (M.G.); jguarinos@gmail.com (J.G.); abardaji.hj23.ics@gencat.cat (A.B.)

**Keywords:** PCI: percutaneous coronary intervention, CTO: chronic total occlusion, CA: coronary arteries

## Abstract

This study aimed to analyze angiographic characteristics of new attempted percutaneous coronary intervention (PCI) on chronic total occlusion (CTO) compared to first attempt group. The cohort of 527 CTO-PCIs was divided into first-attempt and re-attempt groups, and angiographic characteristics, level of complexity, and contributing factors to failure were analyzed. Between-group success rate difference and potential angiographic and technical aspects contributing to the success in new attempts were scrutinized. A total of 47 new PCIs in 39 patients were performed. The reattempt group showed higher J-CTO score compared to the first-attempt group (2.4 ± 1.06 vs. 1.2 ± 1.06; *p* < 0.001). The use of more complex techniques and devices such as retrograde approach (29.8% vs. 12.9%) and IVUS (48.9 vs. 27.3%; *p*: 0.002) were more frequent in the reattempt group. Both procedural and fluoroscopy time were higher in the reattempt group (197 ± 83.9 vs. 150.1 ± 72.3 and 97.7 ± 55.4 vs. 68.7 ± 43, respectively; *p* < 0.001). There was no between-group difference in terms of technical success (79.8 vs. 76.6% for first attempt vs. reattempt group, respectively; *p*: 0.6). The overall success rate increased by 6.1%, achieving 85.9% in the entire cohort. Reattempted CTO-PCIs required more complex techniques and had comparable technical success rate with regard to the first-attempt group.

## 1. Introduction

Although successful recanalization of chronic total occlusions (CTO) has considerably increased over time thanks to improvement in devices and strategies [1], it is still lower than percutaneous coronary intervention (PCI) of other complex non-occluded arteries [2]. It is well-known that more complex CTOs with unfavorable angiographic features are more likely to fail during attempt of recanalization [3]; however, previously unsuccessful procedures can be tried again in repeated interventions using more complex strategies and/or by more experienced operators in order to gradually increase overall CTO-PCI success [4].

Despite the fact that failed CTO-PCIs are allegedly more complex from an angiographic and anatomical point of view, the information about angiographic characteristics and modifiable contributors for failure are scarce in literature. Furthermore, identification of those factors which have contributed to the failure in the first attempt and the key elements and strategies which can potentially conduct to increase the success rate in reattempted CTO-PCIs is paramount for operators while dealing with a previous failed CTO [5].

We aimed to study angiographic and anatomical features of the subgroup of repeated CTO-PCIs in the entire cohort of CTO interventions in our institution. In addition, we tried to identify all potential factors that could have contributed to the final success in the reattempt PCI group.

## 2. Methods

All information regarding CTO-PCIs performed in our institution have been introduced into a database since the installation of the program in May 2007. As part of local operators’ training, five consecutive workshops with experienced Japanese operators in the field of CTO-PCI were organized in our institution over five consecutive years. The main part of the interventions was performed by one operator, although a few cases in the first 100 PCI block were carried out by a second local operator, and a small number of PCIs were done by seasoned operators in the above-mentioned workshops. CTO was defined as the presence of thrombolysis in myocardial infarction (TIMI) flow 0 within an occluded artery standing for more than 3 months [6]. The technical success was defined when CTO body was successfully crossed by a guidewire and device and a TIMI III final flow was achieved after balloon dilatation and stent implantation with less than 30% of residual lesion. The level of CTO difficulty was assessed by J-CTO score and variable assignment for every case was carried out by two observers working in our cath lab. In the event of any inter-observer discrepancy, the opinion of a third examiner was requested, and a final consensus was established. J-CTO variables of 50 randomly selected cases were examined again, and the level of concordance between two observers was estimated in order to assess and solve any possible inter-observer bias. In-hospital major adverse cardiac and cerebrovascular events (MACCEs) including in-hospital death, peri-procedural myocardial infarction, stroke, coronary perforation requiring pericardiocentesis, and major vascular complication needing either percutaneous or surgical intervention were registered.

Antegrade approach was the main strategy in the initial steps of our learning curve, and retrograde access was later incorporated as the level of experience increased over time both after failure of antegrade or as an initial plan.

The indication for recanalization of a CTO was based on the demonstration of ischemia and/or viability in myocardium territory irrigated by the occluded artery using myocardial perfusion scan or cardiac magnetic resonance. A chance to reattempt a CTO in previous failed cases was given to the patient and referring physician, and the procedure was repeated at the earliest time, 4 weeks after the last intervention with the objective to allow the healing of any possible dissection due to the first attempt. The cohort of CTO-PCIs was divided into two groups, the first-attempt PCIs and the reattempt block. A comparison of angiographic characteristics, lesion complexity, and technical success rate between groups was performed. A thorough review of repeated procedures was carried out and compared with the previous failed interventions in search of any potential angiographic changes from the first to the new attempt such as more visibility of CTO body or change in the CTO length. Furthermore, those potential factors contributing to the success of reattempted cases such as use of new techniques or devices, the application of new strategies, and the involvement of more experienced operators in the second attempt were carefully analyzed in this study. Each patient gave written and informed consent before undergoing PCI.

## 3. Statistical Analysis

Continuous data were described as mean ± SD and compared using Student’s *t*-test or Wilcoxon’s rank sum test, as appropriate. Categorical variables were expressed as percentages and were compared using *X*^2^ test or Fisher’s exact test as appropriate. The level of inter-observer concordance regarding the measurement of J-CTO variables was assessed by Kappa index statistic. T-test pairs statistic was used in order to identify any significant changes in CTO length from the first attempt to the second. All analyses were performed using the statistical package of SPSS version 19.0 (IBM Corp., Armonk, NY, USA). The *p*-value of *p* < 0.05 was considered statistically significant.

## 4. Results

A total of 527 of CTO-PCIs in 445 patients between May 2007 and November 2020 were performed in our cath lab. The mean age of the entire cohort was 65.2 ± 10.9 years, and 446 (84.6%) were men. During the first period up to December 2012, the load of cases was low, with a total of 123 PCIs and from January 2013 to November 2020, and an average of 50 cases a year were performed in our institution. The level of inter-observer concordance for the estimation of J-CTO variables assessed by Kappa index was 0.8. A flowchart of the entire cohort comprising of the first-attempt and reattempt groups with the corresponding technical success is depicted in Figure 1. A total of 39 patients had repeated attempts on the same artery after failing the first procedure. In total, 1 out of 39 patients had two CTOs, each of them attempted twice. One patient had two CTOs, one of them was attempted twice and the other four times, and one patient had one CTO that was tried four times. The remaining patients had one CTO attempted twice. The overall repeated procedures were 47 PCIs. We found that 36 out of 47 (76.6%) reattempted CTO-PCIs were successful, resulting in an overall PCI success rate per patient and artery of 85.9% in the entire cohort. This signified that the overall success (taking into account the first and new attempted procedures) increased by 6.1% in the entire cohort.

Procedural success on reattempted CTO increased over time from 65.2% in the first block of 200 first CTO-PCIs to 87.5% in the next block of 201–527 cases, despite the higher level of complexity in the second compared to the first period (70% of cases with J-CTO ≥ 3 in the second period compared to 13% in the first block).

Analysis of CTO-PCI-failure mode in the entire cohort revealed inability to cross the wire through CTO body in 85 (78.7%), device uncrossable lesion in 6 (5.6%), inadequate guiding catheter support in 5 (4.6%), suboptimal TIMI final flow in 2 (1.9%), and patient’s intolerance or development of complications during the intervention in 10 (9.3%).

A comparison of basal and demographic characteristics between first-attempt PCIs and the reattempt group is shown in Table 1. The reattempt group presented a higher rate of patients with previous CABG (14.9 vs 7.3%) with a trend towards significance (*p* = 0.066). Multivessel disease patients were similarly distributed in both groups (64.3% for first-attempt vs. 55.3% for new attempted group; *p*: 0.22). The first attempt group showed left ventricular ejection fraction more frequently < 40% (16 vs. 0%; *p*: 0.004). Between-groups comparison of angiographic characteristics is depicted in Table 2. The main vessel treated in both groups was the right coronary artery (RCA; 44.4% for the first attempt and 63.8% for the reattempt group). In terms of level of complexity, the reattempt CTO-PCI group showed a statistically higher J-CTO score (2.4 ± 1.06 vs. 1.2 ± 1.06; *p* < 0.001).

Technical approach and procedural outcomes are shown in Table 3. Retrograde approach was more frequently used in the reattempt group (29.8 vs. 12.9%) with statistical difference (*p* < 0.005) between groups regarding different technical modalities. IVUS was used more frequently in the new attempt group (48.9 vs 27.3%), a difference considered statistically significant (*p*: 0.002).

Both procedural and fluoroscopy time were higher in reattempt than in first-attempt groups (197 ± 83.9 vs 150.1 ± 72.3 and 97.7 ± 55.4 vs 68.7 ± 43 respectively; *p* < 0.001 for both comparisons). There was not a statistical between-group difference with regard to contrast medium usage and complications including in-hospital death.

The analyses of 47 repeated procedures revealed that subintimal plaque modification (SPM) had been used in 19 cases during the previous attempts. In 14 out of 19, the procedure was successful. In 27 cases, a new strategy including new devices or techniques different to the previous procedures was used, and the new PCI was successful in 22 out of 27. Nine new procedures were performed by a new operator, of which seven were successful. Finally, in 10 new PCIs, the CTO route was more visible in comparison to the previous attempt, and in 9 of 10, the new attempt was successful (Figure 2).

CTO body length mean was reduced from 18.91 to 16.67 mm from the first to the second attempt (mean paired difference: 2.24, CI: 0.43–4.07; *p*: 0.017; correlation coefficient: 0.93; *p*: 0.001). Comparison of technical success between first-attempt and new attempt groups was not statistically significant (79.8% vs. 76.6%, respectively; *p*: 0.6).

Details of procedural approach and outcomes of reattempted CTO-PCIs according to different strategies are depicted in Table 4.

## 5. Discussion

The main findings of this study were that reattempted CTO-PCIs have a comparable procedural success rate with the first attempt interventions, despite the fact that they showed a higher level of complexity in terms of J-CTO score. In this series, repeated procedures on previously failed CTO-PCIs contributed to an increase of overall success rate by 6.1%. The combination of new strategies and different devices, the involvement of new operators, and the appearance of various favorable angiographic changes such as more visibility of CTO body and CTO length reduction in reattempted procedures may have contributed to achieving a comparable success rate with the first-attempt group despite a higher level of complexity in the reattempt group.

CTO-PCIs compared to interventions on non-occluded arteries are technically more challenging, more time-consuming, and have a lower success rate [7]. The operators need specific training and to become accustomed to managing dedicated devices and techniques to overcome different obstacles they could encounter during the intervention [8]. Moreover, patients undergoing a CTO-PCI, especially with higher complexity features such as severe calcification, longer length, and proximal cap ambiguity, are more likely to be exposed to a higher level of radiation [9], which is sometimes a condition for having to interrupt the procedure. Therefore, some of the complex CTOs could be retried in case of failure in the first attempt after investing a reasonable amount of time and contrast medium, especially if no substantial progress has been achieved. There are some algorithms that provide specific guidance for operators to consider stopping the procedure such as when procedure time exceeds 3 h, if more than 3.7 mL x estimated glomerular filtration rate of contrast has been used, and if the radiation dose was more than 5 Gy air Kerma unless the procedure is considerably advanced [10].

The understanding of failure mode and identification of the modifier contributors to failure is extremely important since those factors may be addressed in a new attempt so as to facilitate the future procedural success. A thorough angiographic review is a key component in identifying collaterals, microchannels, proximal cap ambiguity, and all other factors that determine the type of strategy when planning a CTO-PCI [11]. Although the most common cause of procedure failure in our series is similar to those reported in the literature [12], we identified some other factors that were relatively easy to modify such as lack of an adequate guiding catheter support in five (4.6%) of the previous failed cases.

Although a CTO-PCI failure in a dedicated session can be disappointing for the operator, the proximal cap modification as a consequence of attempting to cross the lesion or due to balloon dilatation can facilitate the way for a second attempt and may eventually increase the final success rate [13]. This is a bailout strategy known as SPM when the antegrade wire cannot successfully enter into true lumen and balloon dilatation in subintimal space is performed in order to restore some antegrade flow. SPM and even modified SPM with larger balloon to vessel ratio (0.75:1 or 1:1) has been associated with a higher recanalization rate in coronary angiogram control and higher PCI success in reattempted interventions [14]. In our series, we used this bailout technique in 19 (40.4%) cases after failing to introduce the wire into true lumen in the first attempt. In 14 out of 19 (73.7%) cases, the procedure was successful once reattempted. In fact, in 10 out of 47 (21.3%) in our series, the CTO route was more visible when the procedure was repeated, which indicated some modification in proximal cap and CTO body as a consequence of manipulation with device and/or balloon dilatation during the first attempt.

Our results coincide with those of the Japanese series (Analysis of Japanese CTO-PCI Expert Registry) [4] in terms of higher mean J-CTO score in reattempt CTO-PCI (2.4 ± 1.06 vs. 1.2 ± 1.06 in our series and 2.86 ± 1.03 vs. 1.68 ± 1.05 in the Japanese series) but differ with regard to lower technical success in reattempt group than first attempt group in the Japanese series. This is probably owing to the fact that the Japanese series are represented by more complex cases.

Karacsonyi et al. [15] analyzed and compared outcomes of no previous failed CTO-PCI attempt (1017) with previous failed CTO-PCI attempt (215) in 12 American centers and concluded again that the reattempt group had a higher J-CTO score but found similar between-group technical success and higher procedure and fluoroscopy time as well as higher rate of retrograde approach in the reattempt group, which coincides with our study.

The comparable CTO-PCI success rate between first attempt and reattempt group is in contradiction to the j-CTO stratification, which provides one negative predictive point to reattempted CTOs. This paradoxical situation (“paradox of second attempt”) can have several explanations. Firstly, the first attempted CTO-PCI can serve as an investment procedure as a final resort in order to facilitate the success rate in a new attempt as we explained previously [16]. Secondly, the success rate of a CTO-PCI is highly operator-dependent, and at times seasoned operators can achieve a high success rate, even in highly complex CTOs [17]. The operator’s expertise is a crucial factor along with anatomical variables for a procedural success. Indeed, a new scoring system proposed by our group (E-CTO score) to predict the procedural success comprises the operator’ expertise in a cohort characterized by growing experience in the field of CTO-PCI [18]. Finally, as it has been evidenced in other studies, the operators invest more time and use more complex techniques in reattempted procedures since the meticulous review of previous failed cases can highlight the design of a better strategy and techniques not necessarily used in the first attempt such as retrograde approach, reverse CART, and IVUS guided for antegrade wire re-introduction into true lumen [19,20]. This latter technique consists of positioning the IVUS probe in false lumen, and re-introduction of a second wire into true lumen guided by IVUS was used in two reattempted CTO-PCIs in our series, which revealed again the use of more complex techniques in repeated PCI group (Figure 3).

## 6. Limitation

The main limitation of this study is the small sample of the cohort that impedes the establishment of a plausible relationship between potential angiographic and technical factors with PCI’s final results with statistical power. Furthermore, the results of this study cannot be extrapolated to other centers as these kinds of interventions are highly operator-dependent. Another limitation of this study is the fact that despite the failed cases being given the opportunity to have a new intervention, 50 out of 97 failed cases were not retried, and the results probably would have changed if all failed cases had been reattempted.

## 7. Conclusions

Reattempted CTO-PCIs result in a comparable success rate to first attempt procedures, despite the fact that they presented a higher level of complexity. Repeated CTO-PCI increased the overall success rate by 6.1% in our series. More complex techniques and strategies were used in reattempt procedures that were associated with a longer procedure and fluoroscopy time than first attempt interventions. Some angiographic changes from the first to second intervention, the use of different techniques and strategies in new attempts, and previous investment procedure may have contributed to the similar success rate in reattempted PCIs with regard to that of first-attempt interventions.

## Figures and Tables

**Figure 1 jcm-10-05661-f001:**
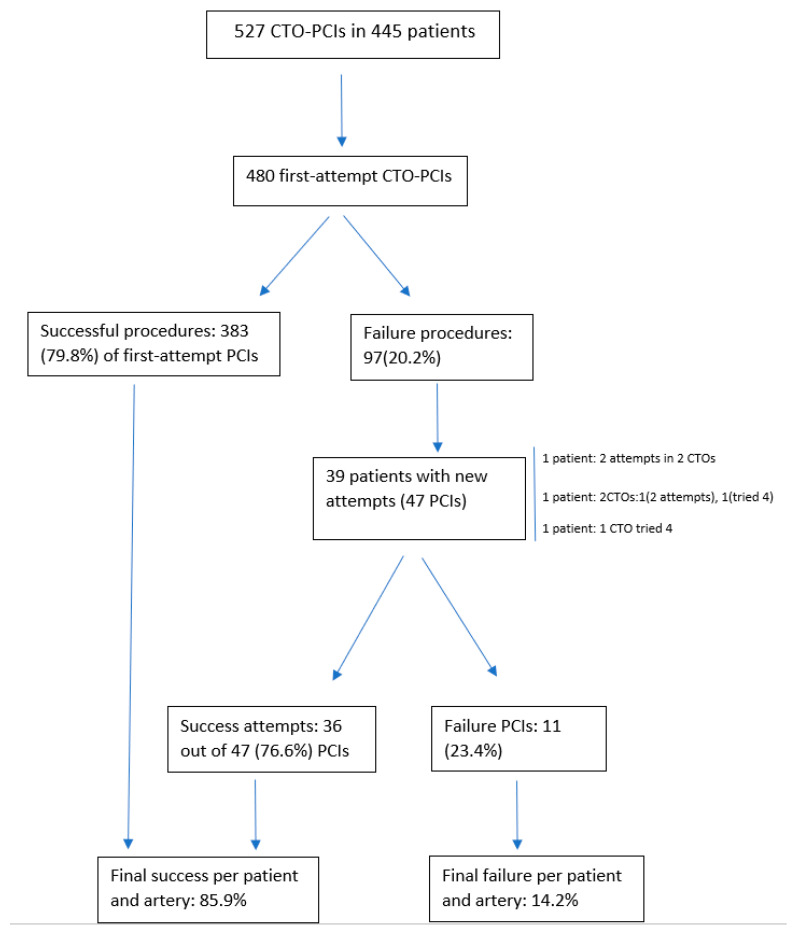
Flowchart of the entire cohort including new attempts.

**Figure 2 jcm-10-05661-f002:**
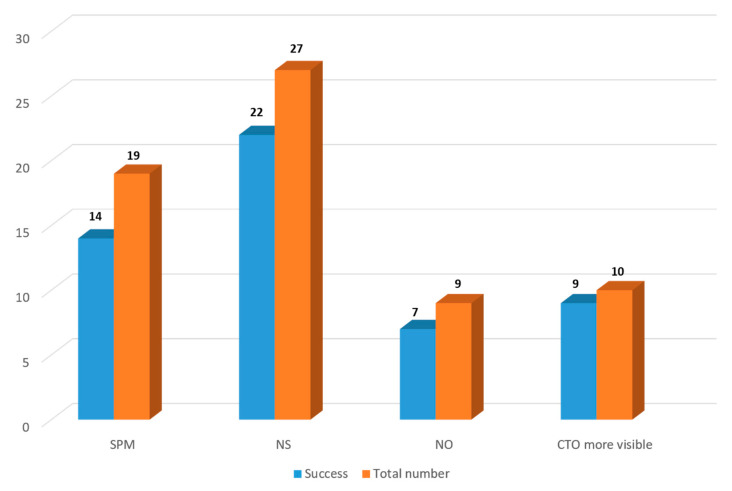
CTO-PCI success rate in different subgroups: SPM in the previous attempt, the use of new strategy, procedure performed by a new operator and based on more visibility of CTO body in the new attempt. CTO: chronic total occlusion; NO: new operator; NS: new strategy; SPM: subintimal plaque modification (during the previous failed attempt).

**Figure 3 jcm-10-05661-f003:**
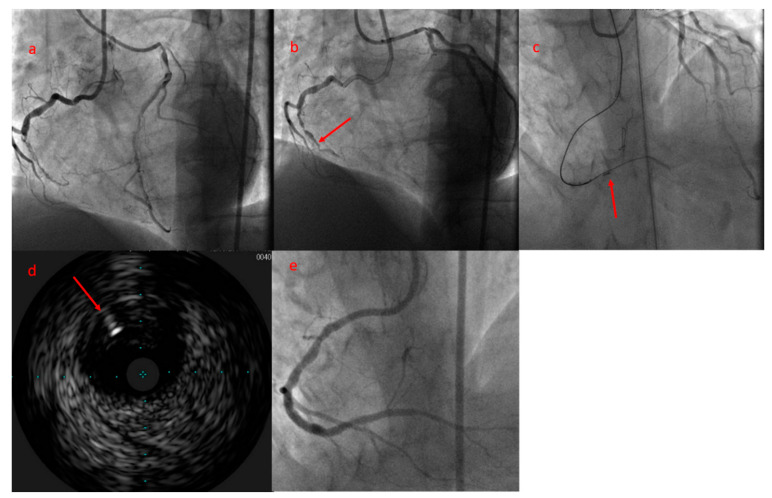
(**a**) Bilateral injection shows an RCA-CTO with 47 mm length. During the first attempt, balloon dilatation in CTO body as a bailout technique was used. (**b**) Coronary angiogram during the second attempt shows CTO body more visible, at least in its first segment (arrowhead), and the CTO length was reduced to 28 mm as a consequence of SPM in the previous procedure. (**c**) The antegrade wire navigated subintimally, and by positioning the IVUS probe in subintimal space (arrowhead), a new wire penetrated into true lumen. (**d**) IVUS shows that the true lumen is compressed between 9 and 12 o’clock, and a new dedicated wire is being penetrated into true lumen (arrowhead). (**e**) Final angiographic result. CTO: chronic total occlusion; RCA: right coronary artery; SPM: subintimal plaque modification.

**Table 1 jcm-10-05661-t001:** Basal and demographic characteristics comparison of first-attempt and reattempted CTO-PCI in the study population.

	First-Attempt PCI (*n* = 480)	Reattempted PCI (*n* = 47)	*p*-Value
Age	65.4 ± 10.97	62.8 ± 9.9	0.11
Male	404 (84.2)	42 (89.4)	0.34
Hypertension	358 (74.6)	38 (80.9)	0.34
Diabetes	216 (45)	23 (48.9)	0.6
Prior MI	219 (45.6)	20 (42.6)	0.69
Prior CABG	35 (7.3)	7 (14.9)	0.066
MVD	308 (64.3)	26 (55.3)	0.22
EF < 40%	77 (16%)	0 (0)	0.004

CABG: coronary artery bypass graft; EF: ejection fraction; MI: myocardial infarction; MVD: multivessel disease.

**Table 2 jcm-10-05661-t002:** Angiographic characteristics of first-attempt and reattempted CTO-PCIs.

	First-Attempt PCI *n* = 480	Reattempted PCI *n* = 47	*p*-Value
CTO site			
LAD	165 (34.4)	9 (19.1)	
LCX	89 (18.5)	7 (14.9)	
RCA	213 (44.4)	30 (63.8)	
DG	6 (1.3)	1 (2.1)	0.24
LM	2 (0.4)	0 (0)	
RI	3 (0.6)	0 (0)	
SVG	2 (0.4)	0 (0)	
In-stent CTO	23 (4.8)	2 (4.3)	NS
J-CTO score	1.2 ± 1.06	2.4 ± 1.06	<0.001

CTO: chronic total occlusion; DG: diagonal artery; LAD: left anterior descending coronary artery; LCX: left circumflex artery; LM: left main; NS: not significant; PCI: percutaneous coronary intervention; RI: ramus intermedius artery; SVG: saphenous vein graft.

**Table 3 jcm-10-05661-t003:** Technical approach and procedural outcomes comparison between first attempt and reattempted procedures.

Variable	First-Attempt PCI *n* = 480	Reattempted PCI *n* = 47	*p*-Value
Approach			
Antegrade	412 (85.8)	31 (66)	
Retrograde	62 (12.9)	14 (29.8)	
ADR	3 (0.6)	1 (2.1)	0.005
ADR+retrograde	2 (0.4)	1 (2.1)	
ADR+retrograde+antegrade	1 (0.2)	0 (0)	
Procedure time (min)	150.1 ± 72.3	197 ± 83.9	<0.001
Fluoroscopy time (min)	68.7 ± 43	97.7 ± 55.4	<0.001
Contrast medium	252.9 ± 90.6	271 ± 104.5	0.19
IVUS	131 (27.3)	23 (48.9%)	0.002
Technical success	383 (79.8)	36 (76.6)	0.6
In-hospital death	3 (0.6%)	0 (0)	NS
Perforation	3 (0.6%)	0 (0)	NS
MVC	3 (0.6%)	0 (0)	NS
MI	11 (2.3)	0 (0)	0.6

ADR: ategrade dissection reentry; IVUS: intravascular ultrasound; MI: myocardial infarction; MVC: major vascular complication; PCI: percutaneous coronary intervention.

**Table 4 jcm-10-05661-t004:** Procedural approach and outcome of reattempted CTO-PCI.

Procedural Approach	(*N*: 47)	Procedural Outcome	
Antegrade	31	Success	28
		Rescue IVUS guided	2/28
		Failure	3
Retrograde	14	Success	7
		Failure	7
ADR	1	Success	1
ADR + retrograde	1	Success	0

ADR: antegrade dissection reentry; IVUS: intravascular ultrasound.

## Data Availability

The majority of the data presented in this study are available in the table and figures. Any additional data presented in this study are available on request from the corresponding author.

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
