# Peer review of "Angiographic Characteristics and Outcomes of Percutaneous Coronary Intervention of Reattempted Chronic Total Occlusion: Potential Contributing Factors to Procedural Success"

_jcm, 2021, doi:10.3390/jcm10235661_

Round 1
Reviewer 1 Report
Thank you for the opportunity to review your manuscript.
The authors describe success rates and characteristics of 527 initial and 47 re-attempt CTO PCI lesions at their center from 2007-2020. I appreciate the authors highlighting the limitations of their analysis (in particular the single-center nature and low number of cases, especially re-do cases). I further appreciated the review of modes of failure as well as complication rates and types, the detailed lesion and procedure analysis for new and re-do cases, the discussion regarding the importance of re-staging at the time of re-attempt, the discussion regarding investment procedures, and the discussion regarding a programmatic approach to CTO PCI as a center-wide team.
A few comments to consider:
It may be worthwhile to specifically note that predictors of success or failure can be broken down not just by patient-related anatomic characteristics but also by operator and institution level characteristics (e.g. individual-operator and center annual and lifetime volume and experience).
In the Results section you mention the higher annual CTO case volume from 2013-2020 compared to 2007-2013. Did the complexity and/or success rates of new attempted cases increase over time as well as the center and team gained more experience over time?
I would also consider a brief discussion regarding the decision not to re-attempt CTO PCI (per your analysis 50 of the 97 failed procedures were not reattempted but there is no mention of why).
Thank you again.
Author Response
Short title has been eliminated
Dear reviewers: we highly appreciate your suggestions after revising our manuscript. Here you have the proper corrections:
It may be worthwhile to specifically note that predictors of success or failure can be broken down not just by patient-related anatomic characteristics but also by operator and institution level characteristics (e.g. individual-operator and center annual and lifetime volume and experience).
In the discussion section you find the following explanation. Besides, we made an additional comment and bibliographic citation (18) based on our previously reported new scoring model (E-CTO) which includes the operator experience as an influential factor for the procedural success along with anatomical variables.
(Secondly, the success rate of a CTO-PCI is highly operator- dependent and at times seasoned operators can achieve high success rate even in high complex CTOs. [17] The operator’s expertise is a crucial factor along with anatomical variables for a procedural success. Indeed, a new scoring system proposed by our group (E-CTO score) for predicting the the procedural success comprises the operator’ expertise in a cohort characterized by growing experience in the field of CTO-PCI. [18])
In the Results section you mention the higher annual CTO case volume from 2013-2020 compared to 2007-2013. Did the complexity and/or success rates of new attempted cases increase over time as well as the center and team gained more experience over time?
We add this comment and data in the result section:
Procedural success on reattempted CTO increased over time from 65.2% in the first block of 200 first CTO-PCIs to 87.5% in the next block of 201-527 cases despite the higher level of complexity in the second compared to the first period (70% of cases with J-CTO ≥ 3 in the second period compared to 13% in the first block).
I would also consider a brief discussion regarding the decision not to re-attempt CTO PCI (per your analysis 50 of the 97 failed procedures were not reattempted but there is no mention of why).
In method section we made a comment about offering the chance to repeat CTO-PCI in failed cases to the patients and their physicians:
A chance to reattempt a CTO in previous failed cases was given to the patient and referring physician and the procedure was repeated at the earliest, 4 weeks…..
Besides we added an observation in this regard in the limitation section:
Another limitation of this study is the fact that despite the failed cases were given the opportunity to have a new intervention, 50 out of 97 failed cases were not retried and the results probably would have changed if all failed cases had been reattempted.

Reviewer 2 Report
The submitted manuscript deals with angiographic characteristics and outcomes of percutaneous coronary intervention of reattempted chronic total occlusion. CTO are a very interesting field of coronary cardiovascular intervention, that is ever expanding due to new dedicated materials end technique. The manuscript is well written, even if with some minor spelling errors that I encourage the authors to correct. However, this Reviewer has some request:
- Authors declared that "The analyses of 47 repeated procedures revealed that subintimal plaque modification (SPM) had been used in 19 cases during the previous attempts". J-CTO score is obviously higher in reattempted CTO due to the presence of the item "repeated attempt" in the score, but authors demonstrated that SPM during first procedure can help procedural success in the second procedure. Can the authors try to define the relationship between SPM and procedural success of the second procedure? I.E: multivariable logistic regression analysis of J-CTO score items plus SPM and procedural success of the second procedure.
- In some table p value are missing for some items or declared as NS: please provide numerical data for all variables.
Author Response
Dear reviewers: we highly appreciate your suggestions after revising our manuscript. Here you have the proper corrections:
It may be worthwhile to specifically note that predictors of success or failure can be broken down not just by patient-related anatomic characteristics but also by operator and institution level characteristics (e.g. individual-operator and center annual and lifetime volume and experience).
In the discussion section you find the following explanation. Besides, we made an additional comment and bibliographic citation (18) based on our previously reported new scoring model (E-CTO) which includes the operator experience as an influential factor for the procedural success along with anatomical variables.
(Secondly, the success rate of a CTO-PCI is highly operator- dependent and at times seasoned operators can achieve high success rate even in high complex CTOs. [17] The operator’s expertise is a crucial factor along with anatomical variables for a procedural success. Indeed, a new scoring system proposed by our group (E-CTO score) for predicting the procedural success comprises the operator’ expertise in a cohort characterized by growing experience in the field of CTO-PCI. [18])
In the Results section you mention the higher annual CTO case volume from 2013-2020 compared to 2007-2013. Did the complexity and/or success rates of new attempted cases increase over time as well as the center and team gained more experience over time?
We add this comment and data in the result section:
Procedural success on reattempted CTO increased over time from 65.2% in the first block of 200 first CTO-PCIs to 87.5% in the next block of 201-527 cases despite the higher level of complexity in the second compared to the first period (70% of cases with J-CTO ≥ 3 in the second period compared to 13% in the first block).
I would also consider a brief discussion regarding the decision not to re-attempt CTO PCI (per your analysis 50 of the 97 failed procedures were not reattempted but there is no mention of why).
In method section we made a comment about offering the chance to repeat CTO-PCI in failed cases to the patients and their physicians:
A chance to reattempt a CTO in previous failed cases was given to the patient and referring physician and the procedure was repeated at the earliest, 4 weeks…..
Besides we added an observation in this regard in the limitation section:
Another limitation of this study is the fact that despite the failed cases were given the opportunity to have a new intervention, 50 out of 97 failed cases were not retried and the results probably would have changed if all failed cases had been reattempted.
Shor title has been eliminated.
We added the following sentence into the method section:
All patients signed informed consent before undergoing the procedure.
